A novel few-shot learning based multi-modality fusion model for COVID-19 rumor detection from online social media

Lu Heng-yang luhengyang@jiangnan.edu.cn 1 2
Fan Chenyou 3
Song Xiaoning 1
Fang Wei 1
1 Jiangsu Provincial Engineering Laboratory of Pattern Recognition and Computational Intelligence, School of Artificial Intelligence and Computer Science, Jiangnan University , Wuxi , China
2 State Key Laboratory for Novel Software Technology, Nanjing University , Nanjing , China
3 Shenzhen Institute of Artificial Intelligence and Robotics for Society , Shenzhen , China
Zubiaga Arkaitz
Electronic publication date: 2021 Aug 20
Publication date: 2021
Volume: 7
Electronic Location ID: e688
Received 2021 Mar 31; Accepted 2021 Aug 3
Copyright: ©2021 Lu et al.
Copyright year: 2021
Copyright holder: Lu et al.
License: This is an open access article distributed under the terms of the Creative Commons Attribution License, which permits unrestricted use, distribution, reproduction and adaptation in any medium and for any purpose provided that it is properly attributed. For attribution, the original author(s), title, publication source (PeerJ Computer Science) and either DOI or URL of the article must be cited.
License URL: https://creativecommons.org/licenses/by/4.0/

Keywords: Rumor detection, Few-shot learning, Social media, COVID-19, Multi-modality

Funding: National Natural Science Foundation of China 62002137 61876072 Fundamental Research Funds for the Central Universities JUSRP12021 State Key Lab Novel Software Technology, Nanjing University, P.R. China KFKT2020B02 This work was supported by the National Natural Science Foundation of China (No.62002137, 61876072), the Fundamental Research Funds for the Central Universities (No. JUSRP12021), and the State Key Lab for Novel Software Technology, Nanjing University, P.R. China (No. KFKT2020B02). The funders had no role in study design, data collection and analysis, decision to publish, or preparation of the manuscript.

==============================
Background

Rumor detection is a popular research topic in natural language processing and data mining. Since the outbreak of COVID-19, related rumors have been widely posted and spread on online social media, which have seriously affected people’s daily lives, national economy, social stability, etc. It is both theoretically and practically essential to detect and refute COVID-19 rumors fast and effectively. As COVID-19 was an emergent event that was outbreaking drastically, the related rumor instances were very scarce and distinct at its early stage. This makes the detection task a typical few-shot learning problem. However, traditional rumor detection techniques focused on detecting existed events with enough training instances, so that they fail to detect emergent events such as COVID-19. Therefore, developing a new few-shot rumor detection framework has become critical and emergent to prevent outbreaking rumors at early stages.

Methods

This article focuses on few-shot rumor detection, especially for detecting COVID-19 rumors from Sina Weibo with only a minimal number of labeled instances. We contribute a Sina Weibo COVID-19 rumor dataset for few-shot rumor detection and propose a few-shot learning-based multi-modality fusion model for few-shot rumor detection. A full microblog consists of the source post and corresponding comments, which are considered as two modalities and fused with the meta-learning methods.

Results

Experiments of few-shot rumor detection on the collected Weibo dataset and the PHEME public dataset have shown significant improvement and generality of the proposed model.

Introduction

From the early social psychology literature, a rumor refers to a story or a statement whose truth value is unverified or deliberately false (Allport & Postman, 1947). More recently, DiFonzo & Bordia (2011) defined rumor as unverified and instrumentally relevant information statements in circulation that arise in contexts of ambiguity and that function primarily to help people make sense and manage threat. With the fast development of the Internet, the widespread of rumors online has become a major social problem nowadays. Especially on popular online social media such as Sina Weibo and Twitter, users or machines post millions of unverified messages every day. Since the breakout of COVID-19, rumors about COVID-19 have been continuously posted and spread, causing the panic of the public and placing considerable losses on the economy and other aspects of society. Thus, the study of discovering and dispelling rumors fast and accurately has become both theoretically and practically valuable. Therefore, rumor detection on social media has become one of the recently popular research areas.

Online social media are naturally suitable for stimulating mass discussions and spreading information. Users usually initialize conversations over spotlighted events/topics and thus generate a series of related posts over the same events/topics. Each conversation/discussion consists of a source post, corresponding replies and reposts. Therefore, most existing works detect rumors on social media at a macro level. They aim to determine whether the public discussions relating to a certain event/topic belongs to rumor or not (Wu, Yang & Zhu, 2015). Existing works under this setting contain both traditional machine learning models with hand-crafted features (Castillo, Mendoza & Poblete, 2011; Yang et al., 2012; Kwon et al., 2013; Jin et al., 2014; Wu, Yang & Zhu, 2015), and deep learning-based models (Ma et al., 2016; Yu et al., 2017; Chen et al., 2018; Ma, Gao & Wong, 2019; Bian et al., 2020). One of the other research lines aims to detect rumors at a micro level, which means to detect whether a single post belongs to a rumor. It has practical value for those who care more about the credibility of single posts. Pioneer works have been conducted on the Twitter rumor detection task (Sicilia et al., 2017; Sicilia et al., 2018a)). Most existing rumor detection models assume that each event has plenty of training instances and regard the task of rumor detection as the classification problem based on supervised learning. Therefore a coherent challenge of existing rumor detection methods is identifying rumors relating to some suddenly happened events such that very few instances were available at the early stages of the events. For the macro-level models, it is a possible solution to set time windows for learning good features (Kwon, Cha & Jung, 2017) at the early stage. It is still based on supervised learning and the discussed events appear in both the training set and test set.

However, the COVID-19 is an emergent event which has never occurred in the past. This means there is only rarely labeled data for this kind of emergent event at the early stage. In this scenario, all the previous supervised learning-based methods are not applicable, because the training data and the test data belong to distinct events. Previous works on cross-topic rumor detection have discussed this problem, which added knowledge of the test topic in the training set (Sicilia et al., 2018b). According to the conclusion in Sicilia et al. (2018b), to obtain good results in cross-topic detection, at least 80% of the test topic knowledge should be included in the training set. Therefore, existing works have huge difficulty in rumor detection for emergent events like COVID-9 with very little labeled data, the main challenges include: (1) The rumors about the target emergent event to be detected has never occurred before, so that the history data of other events could hardly contribute to building prediction models. (2) The number of labeled instances for the target emergent event is extremely scarce, e.g., only one or three or five, which makes the popular “pretraining and finetuning” paradigm fail under this situation. Motivated by the necessity of COVID-19 rumor detection under these real challenges, we formulate it as a few-shot learning task. Few-shot learning is able to learn an adaptable model with only a few labeled data. It can predict rumors about emergent events, which have never occurred in the training set. Considering collecting information like the user profile is both time-consuming and privacy-sensitive, we aim to detect rumors only based on the text contents from online social media. We regard a full microblog consists of two modalities, the source post and the limited number of corresponding comments, and aim to detect whether a full microblog belongs to a rumor. Both modalities are used for building fusion models. To the best of our knowledge, this is the first work tackling the challenge of detecting rumors with very few instances over emergent events and considering the rumor detection task as a few-shot learning task. The main contributions are as follows:

• We collect and contribute a publicly available rumor dataset that is suitable for few-shot learning from Sina Weibo, the largest and most popular online social media in China. This dataset contains 11 COVID-19 irrelevant events and three COVID-19 relevant events, which sums to a total of 3,840 instances, of which 1,975 are rumors and 1,865 are non-rumors.

• We propose the novel problem of few-shot rumor detection on online social media. It aims to detect rumors of emergent events, which have never happened, with only a very small number of labeled instances. The definition of instances considers the characteristics of online social media by containing both source posts and corresponding comments.

• We introduce a few-shot learning-based multi-modality fusion model named COMFUSE for COVID-19 rumor detection, including text embeddings modules with pre-trained BERT model, feature extraction module with multilayer Bi-GRUs, multi-modality feature fusion module with a fusion layer, and meta-learning based few-shot learning paradigm for rumor detection. We perform extensive evaluations on benchmark datasets to show that our model is superior to the state-of-the-art baselines in the few-shot situation, which can detect rumors of emergent events with only a small number of labeled instances.

literature review

This paper focuses on the few-shot rumor detection task on social media for the emergent event like COVID-19, related literature reviews include rumor detection, rumor detection at an early stage, and few-shot learning.

Rumor detection

Most early works on rumor detect extracted hand-crafted features and built classifiers under supervised learning. For example, Castillo, Mendoza & Poblete (2011) constructed features from the message, user profiles and topics to study the credibility of tweets by applying SVM and Naive Bayes. Kwon, Cha & Jung (2017) comprehensively explored the user, structural, linguistic and temporal features in rumor detection tasks. Sicilia et al. (2018a) applied new features such as the likelihood a tweet is retweeted, and the fraction of tweets with URLs to detect health-related rumors. In addition, hand-crafted features such as location-based features (Yang et al., 2012), temporal features (Kwon et al., 2013), topical space features (Jin et al., 2016) and sentimental features (Liu et al., 2015; Mohammad, Sobhani & Kiritchenko, 2017) are also applied. In this stage, traditional machine learning algorithms such as support vector machines (Yang et al., 2012) and decision trees (Castillo, Mendoza & Poblete, 2011; Zhao, Resnick & Mei, 2015) were the common choices. However, hand-crafted feature engineering is time-consuming and with high labor costs. Benefit from the development of deep learning, deep-learning based features have been widely applied to rumor detection recently. These features are extracted automatically in the form of embeddings by training deep neural networks. Representative models such as recurrent neural networks (RNNs) and convolutional neural networks (CNNs) are widely used to extract essential features of given texts for rumor detection (Yu et al., 2017; Chen et al., 2018; Ma, Gao & Wong, 2019; Bian et al., 2020). There are also some works considering the characteristics of social media. For the propagation structure in social media, Liu proposed to use the propagation information to help detect rumors (Liu & Xu, 2016). For the response and reply operations in Twitter, retweets (Yuan et al., 2019) and replies (Ma, Gao & Wong, 2019) along with source tweets were utilized.

Early-stage rumor detection

Detect rumors at the early stage is both necessary and challenging. A comprehensive study was conducted to explore rumor detection performance over varying time windows with four kinds of hand-crafted features include user, structural, linguistic and temporal-based features (Kwon, Cha & Jung, 2017). It reveals that user and linguistic features are suitable for building early detection models and proposed a practical algorithm that does not require full snapshots nor complete historical records. Similar strategies were applied to deep learning-based models such as GAN-GRU (Ma, Gao & Wong, 2019) and Bi-GCN (Bian et al., 2020), which set a detection delay time and evaluated with tweets posted no later than the delay. These introduced works detect early rumors at the macro level and the detected events of the discussions online have appeared in both the training set and test set. Another pioneer work of early rumor detection focused on cross-topic rumor detection (Sicilia et al., 2018b), which aims to detect rumors about an unseen topic that has never used and existed in the training set. This paper detected rumors at the micro-level and implies that under this practical setting, it requires at least 80% of the test topic samples to be included in the training set, in order to achieve good results. The cross-topic task was also discussed in a recent proposed work about rumor detection with imbalanced learning (Fard et al., 2020).

Few-shot learning

Few-shot learning assumes that very few labeled instances are available, which is a challenging task in machine learning (Vinyals et al., 2016; Finn, Abbeel & Levine, 2017; Snell, Swersky & Zemel, 2017; Sung et al., 2018). Meta-learning is one of the popular strategies in few-shot learning, developing machine learning models to predict unseen categories with few labeled data. The core idea of meta-learning is to learn transferable knowledge on training data that can adapt to new tasks efficiently with just a few examples of the new tasks. Optimization-based meta-learning approaches such as MAML (Finn, Abbeel & Levine, 2017) aim to search for optimal initial parameters of models which can quickly adapt to new tasks with just a few gradient steps. Meta-transfer learning (MTL) (Sun et al., 2019a; Sun et al., 2019b) proposed to avoid the overfitting problem during training a small amount of data from the unseen category. Metric-based meta-learning approaches such as MatchingNet (Vinyals et al., 2016) and PrototypicalNet (Snell, Swersky & Zemel, 2017) aim to learn a better feature space to reflect the distance between instances. Although few-shot learning has achieved success in image classification tasks, very few research attempts have been made to study how to detect rumors with few instances.

Materials & Methods

Problem setting

This paper models the COVID-19 rumor detection problem as a few-shot binary classification task, denoted as N-way M-event K-shot Q-query. N refers to the distinct number of few-shot learning labels, which we have N = 2 in this paper as we consider an instance as rumor or non-rumor. M represents the number of sampled events among E. Let E = Ep∪Es denote a set of given events, where Ep refers to those events that happened in the past and have enough labeled instances for training, Es refers to those events that happened suddenly and should be predicted with only a small number of labeled instances. K represents the number of sampled instances in the support set (training set) for each label, and Q represents the number of sampled instances in the query set (test set) for each label.

Each event is composed of a set of related instances. Given an instance (xi, yi), xi = [mi, ci], where xi is a full microblog, mi refers to the text content (post) of the ith microblog, and ci = [ci1, ci2, …, cil] consists of the l comments of the ith microblog. We regard mi and ci as two modalities. yi is the label of the ith instance, which indicates whether the ith instance belongs to rumor or not.

The few-shot learning target is to train a classifier ℂ to predict whether an instance in Es belongs to a rumor with only a few numbers of labeled data. Models trained on instances of Ep are used for task adaptation.

Data

Sina Weibo is a popular Chinese online social media platform, where users can post or repost, and leave comments with each other. Figure 1 is a rumor example from Sina Weibo. It mainly contains the post (similar to the source post on Twitter) and corresponding comments (similar to the replies on Twitter). If it is judged as a rumor by the official platform, there is a reminder display on the top of the page.

Figure 1 Example of the Sina Weibo page, which contains a rumor microblog.

We construct and share a novel dataset based on Weibo for the research of few-shot rumor detection (https://github.com/jncsnlp/Sina-Weibo-Rumors-for-few-shot-learning-research). The publicly available dataset is written in Chinese and each instance contains a source post along with corresponding comments, the posted date and its label are also included. Our collected dataset contains 11 independent and distinct COVID-19 irrelevant events that happened in the past and three COVID-19 relevant events that happened since the breakout of COVID-19. For each event, we crawl related microblogs consisting of source posts (modality 1) and corresponding comments (modality 2) from Sina Weibo, in which both rumors and non-rumors are covered. The event names are used as searching keywords. We provide the corresponding descriptions of all events are as follows (the original names are in Chinese, here we have translated them to English).

• MH370: This event is about the crash of Malaysia Airlines MH370 discussed online.

• College entrance exams: This event is about the annual Chinese college entrance exams.

• Olympics: This event is the discussion about the news of Olympics games on Sina Weibo.

• Urban managers: In China an urban manager is someone who helps keep the city clean and safe. This event is the discussion about how urban managements perform their official duties.

• Cola: This event is about Coke Cola from the perspectives of food additives.

• Child trafficking: This event is about child trafficking and asking for help reported on Sina Weibo.

• Waste oil: This event is about the news of waste oil used for cooking from the perspectives of food safety.

• Accident: This event is about accidents that happened and reported on Sina Weibo, such as traffic accidents.

• Earthquake: This event is about the earthquake discussed and reported on Sina Weibo.

• Typhoon: This event is about the typhoon discussed and reported on Sina Weibo.

• Rabies: This event is the discussion about serious death caused by rabies on Sina Weibo.

• Lockdown the city: This event is the discussions about the lock-down-city policy online.

• Zhong Nanshan: This event is about the Chinese anti-epidemic expert Dr. Zhong Nanshan.

• Wuhan: This event is about discussions on the COVID-19 in Wuhan.

The official Sina Weibo community management center (https://service.account.weibo.com/) displays all the fake posts judged and labeled by professional human moderators, which is commonly used as the source of collecting Weibo rumors (Ma et al., 2016; Yuan et al., 2019). Figure 2 illustrated the workflow of the judgement for the rumor displayed in Fig. 1, similar to the process of the court ruling. The final judgement by the official platform (on the top of Fig. 2) comes from both reported reasons from other users (on the bottom left) and explanations from the posted user (on the bottom right). Once the post is labeled as a rumor, a “Fake post” (the original one is in Chinese) sign would appear on the posted page, as Fig. 1 shows. We implement a web crawler to collect all the reported posts from the official Sina Weibo community management center, posted date starts from May 2012 to December 2020. Keywords of distinct events (original formats are in Chinese, translated to English in Table 1) are then applied to filter event-related instances as rumors. To collect non-rumors, we choose the same keywords used for collecting rumors of selected events. The web crawler is designed to search and crawl the posts with given keywords. For those crawled posts which are not marked as “Fake post” by the official platform, we take them as non-rumors. All the corresponding comments are crawled together.

Figure 2 Workflow of the rumor judgement by the official Sina Weibo community management center.

Table 1 Statistics of events for the COVID-19 rumor dataset after removing duplicates.

Event	Rumor	Non-rumor	
		Modality 1	Modality 2	Modality 1	Modality 2	
COVID-19 irrelevant	MH370	134	239	263	156	
College entrance exams	591	945	148	153	
Olympics	82	199	174	149	
Urban managers	150	305	95	71	
Cola	420	407	216	285	
Child trafficking	173	258	95	54	
Waste oil	58	91	134	122	
Accident	83	168	101	77	
Earthquake	59	133	118	85	
Typhoon	65	163	108	90	
Rabies	43	77	102	70	
COVID-19 relevant	Lockdown the city	25	59	87	89	
Zhong Nanshan	22	48	56	44	
Wuhan	70	161	168	154	
Total	1,975	3,253	1,865	1,599	

Due to the repost operation in Sina Weibo, which is similar to the re-tweet feature in Twitter, there exist duplications in the originally collected data. We exploit Hamming distance to filter similar or repetitive texts. Specifically, we treat two source posts with hamming distance less than a threshold (e.g., six) as duplicates and just retain one of them in the dataset. After this deduplication operation, the statistics of the Weibo dataset are as Table 1 shows.

Few-shot rumor detection

The general overflow of COMFUSE is as Fig. 3 shows. The input microblogs consist of source posts along with corresponding comments. Firstly, the pre-trained Bidirectional Encoder Representations from Transformers model (BERT) is applied to achieve the word embeddings of the input microblogs. Then two bidirectional GRUs are used to learn features of source posts and comments separately. A fusion layer is applied to fuse the features of both modalities, which are source posts and comments. Finally, meta-learning is applied to detect rumors related to new events with task adaptation.

Figure 3 Workflow of COMFUSE.

Pretrained word embeddings

Recently, transformer-based NLP models (Vaswani et al., 2017) have shown that attention-based embedding mechanisms have great superiority over simple structured embedding models (Sun et al., 2019a; Sun et al., 2019b), such as word2vec (Mikolov et al., 2013) and GloVe (Pennington, Richard & Christopher, 2014). In this paper, we utilize BERT models pre-trained on large-scale such as Wikipedia with Transformers to embed the inputs. Given an instance xi = [mi, ci1, …, cil], both the source posts and comments are in the format of sequences. Figure 4 demonstrates embedding inputs with pre-trained BERT in detail. For an input (a post mi or a comment cil), the first step is tokenization based on the predefined vocabulary and achieve [t1, t2, …, tn], where n is the number of tokens. An embedding layer is then applied to achieve initialized embeddings ek for every token tk and achieve [e1, e2, …, en]. Then, the embeddings B = [b1, b2, …, bn] become the output with transformer models. For the given input xi = [mi, ci1, …, cil], the outputs of this procedure are corresponding pre-trained BERT embeddings, denoted as [Bmi, Bci1, …, Bcil].

Figure 4 Illustrations of word embeddings with BERT.

Bi-GRUs feature extractions

Recently, it is the mainstream to extract features from texts with deep neural networks. Representative RNNs-based models such as LSTMs and GRUs have shown effectiveness in the rumor detection task (Liu, Jin & Shen, 2019; Wang & Guo, 2020). In this paper, we apply bidirectional GRUs (Bi-GRUs) to extract features of source posts and corresponding comments. We take the post m as an example. After applying pretrained BERT embeddings, the input post m turns to the embeddings matrix Bm=b1m,b2m,…,bnm, where n is the same definition of token numbers. The BiGRUs are applied upon the embedding matrix to further decode post m to textual hidden features denoted as Hm=h1m,h2m,…,hnm. The general structure of Bi-GRUs is as Fig. 5 shows.

Figure 5 Structure of Bi-GRUs.

For the jth input embeddings bjm, the decoded features hjm of Bi-GRUs’ outcome in one direction can be denoted as hjm=GRUsbjm,hj−1m. In Eqs. (1)–(4) we show its complete form in detail: (1) rjm=σWrbjm+βr+Whrhj−1m+βhr

(2) zjm=σWzbjm+βz+Whzhj−1m+βhz

(3) hjm′= tanhWhbjm+βh+rjm∗Whhhj−1m+βhh

(4) hjm=1−zjm∗hjm′+zjm∗hj−1m

The hidden states of the forward input sequence with n tokens can be represented as Hm ⃗=GRUsBm ⃗,h0, where h0 is the initial hidden state. Similarly, the hidden states of the backward input sequence are represented as Hm ⃖=GRUsBm ⃖,h0. The final hidden states of both directions are calculated as Eq. (5) shows, which are also regarded as the features for further rumor detection. (5) Hm=Hm ⃗+Hm ⃖2

For the i-th microblog, the extracted feature of the post mi is denoted as Hmi, similarly, the extracted features of comments ci1,ci2,…,cil are denoted as Hci=H1ci,…,Hlci.

Feature fusion layer

Taking the ith instance from Sina Weibo, for example, we consider both source post mi and corresponding comments [ci1, …, cil] as two modalities. Because each instance may have more than one comment, the fusion layer contains two main steps. The first step is to fuse the features H1ci,…,Hlci of all comments, denoted as Hci. The second step is to fuse the features of the source post and comments.

In the first fusion step, the features of all the l comments in the same instance are extracted via the same Bi-GRUs. As the comments of each microblog are embedded into the same feature space, it is natural to fuse them with the weighted sum of their features. We regard the contribution of each comment to be equal for the rumor detection task, which is defined in Eq. (6). (6) Hci= ∑j=1lHjci

In the second fusion step, the features of the source post and comments in the same instance are extracted via two individual Bi-GRUs. Following the common practice, we fuse these multi-modal features together with concatenation to build the features of the microblogs. For the ith instance, the fused feature is defined as Hi = [Hmi; Hci].

Few-shot learning

Usually, rumors from online social media are usually produced according to certain events. Rumors of emergent events could be very distinct from events collected in the past, so that rumor detection models could barely generalize on new events. However, breaking events like COVID-19 are unprecedented so that very rare instances are available. This may result in the failure or overfitting to directly build a rumor detection model based on supervised learning with the lack of labeled training data for emergent events.

To tackle this challenge, we propose a few-shot learning paradigm by learning a generic model with labeled data from past observed events and adapting to unseen events with only a few labeled instances. We propose a meta-learning based strategy in learning the few-shot rumor detection tasks. The core idea is to sample a large number of task combinations in training instances so that the model can learn the transferable knowledge for unseen categories. State-of-the-art methods are optimization-based with the idea of training a good initialized model which could adapt to unseen categories with only a few gradient steps (Finn, Abbeel & Levine, 2017; Sun et al., 2019a; Sun et al., 2019b).

The learning target of few-shot rumor detection with meta-learning methods is to minimize the adaptation loss on unseen tasks during training. Given a batch of few-shot tasks 𝔅 = T1, …, T|𝔅|, the total loss 𝔏 is calculated as Eq. (7) shows. (7) w∗=minwm,wcLwm,wc,s.t.Lwm,wc=1B∑T∈BLTwm,wc

where wm refers to the parameters of the defined Bi-GRUs for dealing with the modality of source posts and wc refers to the parameters of the defined Bi-GRUs for the modality of comments. LT is the loss of task T and w∗ is the optimized model which can fast adapt to unseen events. This optimization problem can be solved iteratively with the steps as shown in Fig. 6, in order to train the models to adapt to sampled new tasks well. We will demonstrate each of the meta-learning steps in detail.

Figure 6 Workflow of one meta-learning iteration.

The COVID-19 rumor detection is defined as the N-way M-event K-shot Q-query few-shot learning task.

Step 1. Sampling: This step aims to sample a few-shot task T from Ep (events happened in the past). Each event has both rumor and non-rumor instances, which means the sampling times from Ep equal to N × M. For an N-way M-event K-shot Q-query few-shot learning task T, K rumor instances and K non-rumor instances are sampled from M events respectively to compose of a support set, denoted as Ts.Q rumor and non-rumor instances are also sampled from the same M events respectively to compose of a query set, denoted as Tq. This step would sample N × M × (K + Q) instances for task T.

Step 2. Adaptation: This step aims to learn latent semantics in unseen categories by adapting the current model to the sampled task T in step 1. This step updates the model parameters wm and wc with the few-shot labeled data in Ts by performing Stochastic Gradient Descent (SGD), as Eqs. (8) and (9) shows. (8) wm′=wm−α∇wmLTswm

(9) wc′=wc−α∇wcLTswc

where α is the step size of adaption, wm′ and wc′ are parameters of the adapted models, which can extract features of source posts and comments in the query set Tq for further rumor detection.

Step 3. Optimization: This step aims to evaluate wm′ and wc′ with more samples in the query set Tq. The empirical loss functions are as Eqs. (10) and (11) show. (10) LTwm=LTqwm′=LTqwm−α∇wmLTswm,

(11) LTwc=LTqwc′=LTqwc−α∇wcLTswc.

To search for the optimal wm and wc defined in Eq. (7), we need to compute the Hessian. However, considering the tradeoff between the computational costs and performance, we solve this problem with just one gradient descent to approximate the parameter updates (Finn, Abbeel & Levine, 2017; Sung et al., 2018). (12) wm←wm−γ∇wmLTwm,

(13) wc←wc−γ∇wcLTwc,

where γ is the learning rate.

To detect rumors about suddenly happened events Es, we can apply the parameters of the adapted models wm′ and wc′ to calculate the probability of the instances with the Sigmoid function.

Experiments and Results

Datasets for experiments

We carry on extensive empirical studies on two real-world datasets with user comments that have been classified as rumors or non-rumors. The first dataset is collected from Sina Weibo, which is used for detecting rumors about COVID-19. The other dataset we use is PHEME (Zubiaga, Liakata & Procter, 2016), which is publicly available and widely used in most rumor detection researches. Details of both datasets are as follows.

Weibo dataset

We collect microblogs that are written in Chinese from Sina Weibo—the largest online social media in China. In this dataset, there are 14 events with 3,840 instances in total. For each event, both rumors and non-rumors are included. Each event is a hot topic such as MH370, COVID-19 expert Zhong Nanshan, etc., which are widely discussed online. Each instance is recorded with a source post along with its comments. To evaluate the performance of few-shot learning on COVID-19 rumor detection, 11 COVID-19 irrelevant events are selected as the training and validation set, and three COVID-19 relevant events are used for testing (listed in Table 1).

PHEME dataset

This is a publicly available dataset (https://figshare.com/articles/PHEME_dataset_of_rumours_and_non-rumours/4010619) with tweets from Twitter in English, which is widely used for the evaluation of rumor detection tasks (Zubiaga, Liakata & Procter, 2017; Ma, Gao & Wong, 2019). It is collected according to five breaking events discussed on Twitter (Zubiaga, Liakata & Procter, 2016). Each instance is recorded with a source tweet along with its reply reactions. To evaluate the performance under the settings of few-shot learning, three breaking events that happened earlier are selected as the training and validation set (#Ferguson unrest, #Ottawa shooting, #Sydney siege), and the remaining two events that happened most recently are used for testing (#Charlie Hebdo shooting, #Germanwings plane crash). The pre-processing of the PHEME dataset follows the practice in previous work (Ma, Gao & Wong, 2019).

For the Weibo dataset, we crawl the comments of microblogs directly as they are readily available on the same webpage with the source posts. For the PHEME dataset, we regard the replies in the given dataset as comments. For the sake of generality, we randomly divide the dataset into training and validation sets according to distinct events, and repeat three times to form three different splits for robust cross-validation. We choose the number of splits as three for the following reasons. In few-shot learning, the number of splits depends on the number of new events in the test set. We take the 2-way 3-event 5-shot 9-query Weibo dataset for example. It has three COVID-19 relevant events to be detected with only a few labeled data. The number of the event in the definition is determined by the number of new events in the test dataset, so it is 3-event. The number of ways indicates the number of labels, which are rumor and non-rumor. With this definition, during the few-shot learning training process, every training epoch will sample three different events in the training set, for each event, five rumor instances and five non-rumor instances will be sampled for training. According to the few-shot learning setting, we guarantee that all events in the training sets should NOT appeared in the testing sets, and vice versa, to avoid the leakage of event information and guarantee that we are testing on complete novel events. We also assume that the number of events in the training set should be no less than the number of events in the test set to ensure the model capacity for adapting to new events. According to our assumption and task settings, we split our Weibo dataset to three events (COVID-19 relevant) for testing, and 11 events (COVID-19 irrelevant) for training. We fix the three events (COVID-19 relevant) for testing, and construct three folds for “cross-validation” over 11 training events (COVID-19 irrelevant) to guarantee that each fold has more than three events in the Weibo dataset. Table 2 displays the statistics of the data for experiments.

Table 2 Statistics of instances in both datasets for experiments.

	Weibo Dataset	PHEME Dataset	
	Training set	Validation set	Test set	Training set	Validation set	Test set	
split 0	1,676	1,736	428	776	889	877	
split 1	2,429	983	889	776	
split 2	2,719	693	995	889	
total number of instances	3,840	2,207	

Baselines for comparisons

Five baselines are selected to compare the performance of few-shot rumor detection, including traditional methods, deep learning methods, and few-shot learning methods.

1. DT-EMB: This baseline model uses the decision tree as the basic classifier, which was applied in traditional rumor detection tasks (Zhao, Resnick & Mei, 2015). The feature of each instance is represented by the embeddings encoded by the same pre-trained BERT model.

2. SEQ-CNNs: This deep learning-based baseline trains classification model with features extracted by CNNs, which is a common choice for rumor detection in recent researches (Yu et al., 2017), the input sequence is encoded by the same BERT pretrained model for fair comparisons.

3. SEQ-Bi-GRUs: This is also a deep learning-based baseline for rumor detection. Bi-GRUs are applied to extract features for training and prediction (Ma et al., 2016; Chen et al., 2018), the input sequence is encoded by the same BERT pretrained model for fair comparisons.

4. GAN-GRU-early: The basic model of this baseline is a popular model named GAN-GRU (Ma, Gao & Wong, 2019). According to the early detection setting in this paper, for each source post, only the latest three comments are used for evaluation, which is as same as modality 2 in COMFUSE.

5. BiGCN-early: The basic model of this baseline is a state-of-the-art model named BiGCN (Bian et al., 2020). According to the early detection setting in this paper, for each source post, only the latest three comments are used for evaluation, which is as same as modality 2 in COMFUSE.

6. COMFUSE-post-only: This is a simplified model of COMFUSE for ablation study. Only the source post of each microblog is used for training and prediction in the few-shot rumor detection task.

7. COMFUSE-com-only: This is another simplified model of COMFUSE for ablation study. Only the comments of each microblog are used for training and prediction in the few-shot rumor detection task.

The problem setting of this paper is few-shot rumor detection, which assumes that the events in the test set have not occurred in the training set and only a small number of labeled instances is available. DT-EMB, SEQ-CNNs, SEQ-Bi-GRUs, GAN-GRU-early and BiGCN-early are five baselines for common rumor detections, which require the training set and test set to share the same events. To have fair comparisons, new paradigms are designed for training and testing these baselines. For the traditional machine learning-based model DT-EMB, a small number of labeled data sampled from the new events are put into the training set for training. For SEQ-CNNs and SEQ-Bi-GRUs, we train the rumor detection model with the training data firstly and finetune the model with a small number of labeled data sampled from the new events. Because the original GAN-GRU and BiGCN are not designed as the few-shot learning models, we use the same instances of new events, which are also used for task adaption in COMFUSE for training. For all models, the same random seed is set for sampling and these sampled data do not appear in the test set.

Experimental settings

According to the problem setting and considering the number of events in the Weibo dataset and PHEME dataset, we define the few-shot rumor detection for the Weibo dataset as 2-way 3-event 5-shot 9-query, for PHEME dataset as 2-way 2-event 5-shot 9-query respectively. We implement COMFUSE with Pytorch 1.8.1 and utilize the pre-trained BERT model from HuggingFace (https://huggingface.co/) to encode the inputs. We use the uncased Chinese model and uncased English model for the Weibo dataset and PHEME dataset respectively. The source code will be publicly available.

To determine the pad size of the input posts and comments, the statistics of the length per text are performed. The histograms of the Weibo and PHEME datasets are as Figs. 7 and 8 show. Considering the trade-off between performance and speed, we set the pad size of posts/source tweets as 100 (for Weibo) and 48 (for PHEME) respectively. We set the pad size of comments/replies as 32 for both datasets. Further experiments are conducted to show the influence of different pad size choices. The experimental results of the Weibo dataset are as Figs. 9 and 10 show.

Figure 7 (A-G) Statistics of length per text of the Weibo dataset.

Figure 8 (A-G) Statistics of length per text of the PHEME dataset.

Figure 9 Experimental results of different pad sizes of source posts with a fixed pad size of comments as 32 on the Weibo dataset.

Figure 10 Experimental results of different pad sizes of comments with a fixed pad size of source posts as 100 on the Weibo dataset.

Figure 9 displays the results of different pad sizes of source posts with a fixed pad size of comments as 32 on the Weibo dataset. Figure 10 displays the results of different pad sizes of comments with a fixed pad size of source posts as 100. Both x-axis refer to the pad size and y-axis refers to the accuracy performance. We can observe that the rumor detection results of COMFUSE with different pad sizes of posts and comments vary slightly. For the Weibo dataset, the experimental results reveal that it is relatively better to set the pad size as 100 for posts and 32 for comments, which is consistent with our decision based on the statistics of the length in Figs. 7 and 8.

In this paper, we define an instance as xi = [mi, ci1, …, cil], which contains l comments. As we consider few-shot learning scenarios, we assume that there are very few useful comments available in the early stage of an event. Thus, in our experiments, we set the number of relevant comments l as 3 for all experiments, in order to simulate the emerging situations and examine whether our approach can successfully detect rumors from very few labeled instances and informative comments.

Performance of few-shot rumor detection

In this paper, we treat the rumor detection task as a binary classification problem, and we use classification accuracy as the evaluation metric for comparisons. We conduct experiments on all three splits and examine their averaged performance, as shown in Tables 3 and 4.

Discussion

Table 3 displays the performance of COVID-19 rumor detection under the few-shot learning setting. A higher classification accuracy indicates a better performance. It can be observed that the traditional machine learning-based method DT-EMB performs poorly in few-shot rumor detection: it achieves only 56.93% accuracy on average, which is barely better than random guessing in a binary classification task.

Two state-of-the-art deep learning-based methods SEQ-CNNs and SEQ-Bi-GRUs achieve similar performance around 68%. They significantly improve the detection over traditional DT-EMB. One of the reasons is the superior ability of deep neural networks to extract important features from contexts, which contribute to the training of models. Furthermore, the paradigm of pretraining first and then finetuning can optimize the model to fit the data of new events to some extent. However, the number of labeled instances of the unseen events for finetuning is quite small in the few-shot rumor detection task, which may result in the underfitting of the fine-tuned model.

GAN-GRU-early and BiGCN-early are another two SOTA baselines and have reported great performance in the traditional rumor detection task, which the events (topics) appear in both the training set and test set. When applying to the emergent rumor detection scenarios, which assumes the events in the test set have never appeared in the training set, the supervised-based GAN-GRU-early and BiGCN-early models show their limitations and are not suitable for the few-shot rumor detection task. One possible reason that GAN-GRU-early underperforms significantly may because there are scarce instances related to the emergent events in the test set being fed to the generators in the training process. This makes the features extracted in the inference pross hardly reflect the instances related to emergent events.

COMFUSE is our proposed multi-modality fusion model for few-shot rumor detection based on the meta-learning approach, with COMFUSE-post-only and COMFUSE-com-only as two simplified versions. COMFUSE-post-only only uses the source posts (source tweets in Twitter) as inputs, as same as DT-EMB, SEQ-CNNs, and SEQ-Bi-GRUs, which are commonly used in existing rumor detection models. Compared with SEQ-CNNs and SEQ-Bi-GRUs, COMFUSE-post-only further improves the few-shot COVID-19 rumor detection accuracy by around 6%. This shows the effectiveness of applying meta-learning methods with only a small number of labeled data to detect rumors of unseen events.

Table 3 The classification accuracy of Weibo dataset in COVID-19 rumor detection.

	split 0	split1	split2	average	
DT-EMB	57.51%	56.82%	56.47%	56.93%	
SEQ-CNNs	66.76%	66.89%	68.48%	67.38%	
SEQ-Bi-GRUs	65.89%	71.09%	69.81%	68.93%	
GAN-GRU-early	60.74%	47.09%	51.45%	53.00%	
BiGCN-early	71.88%	63.46%	63.50	66.28%	
COMFUSE-post-only	73.48%	75.22%	71.17%	73.29%	
COMFUSE-com-only	70.44%	74.59%	74.11%	73.17%	
COMFUSE	79.17%	79.24%	77.41%	78.61%	

Table 4 The classification accuracy of PHEME dataset in latest events rumor detection.

	split 0	split1	split2	average	
DT-EMB	56.92%	56.67%	56.85%	56.81%	
SEQ-CNNs	63.06%	61.42%	63.67%	62.72%	
SEQ-Bi-GRUs	63.25%	60.83%	62.97%	62.35%	
GAN-GRU-early	53.47%	50.43%	56.03%	53.31%	
BiGCN-early	67.94%	57.94%	59.08%	61.65%	
COMFUSE-post-only	63.56%	65.58%	64.17%	64.44%	
COMFUSE-com-only	58.36%	60.47%	57.39%	58.74%	
COMFUSE	68.25%	66.67%	65.39%	66.77%	

COMFUSE takes advantage of both source posts and corresponding comments of the full microblogs to contribute to the detection of rumors from online social media. Intuitively, comments or replies reflect the positive or negative attitudes of the public towards the source posts, so that should provide additional hints towards judging the credibility of the source posts. COMFUSE-com-only is also an ablation model, which only uses the comments of instances for rumor detection. It can be observed that the proposed multi-modality fusion model COMFUSE performs much better than two ablation models COMFUSE-post-only and COMFUSE-com-only, with accuracy improvement by 5%. This shows the necessity of fusing both source posts and comments for rumor detection. Comparing the proposed COMFUSE model with traditional machine learning-based and deep learning-based rumor detection models, it achieves the improvements by 21% and 10% respectively, which shows the superiority of the meta-learning based fusion model for few-shot COVID-19 rumor detection.

Table 4 is the experimental results on the public and commonly used rumor dataset PHEME, to show the generality of COMFUSE. The proposed COMFUSE model also achieved the best performance among all the baselines. As the Weibo dataset has more events and instances available during meta-training, the model can be trained with more and diverse event combinations and thus be more capable of adapting to novel events by learning to capture distinct hints for rumor detection. In contrast, we have to use only 3 events in PHEME training and thus may fail to understand the most distinguishing hints in rumors of the PHEME dataset. This explains why the improvement on the PHEME dataset is not as significant as that on the Weibo dataset.

Conclusions

This paper focuses on the few-shot rumor-detection for unexpected and emergent events which have never or rarely happened before, such as COVID-19. Different from rumor detection on daily events in previous work, emergent events outbreak in sudden so that very few labeled instances can be used for training rumor detection models. As existing rumor detection works assume the events to be predicted are as same as those to be trained, they are greatly limited in rumor detection for emergent events. This paper identifies the rumor detection for emergent events as the few-shot learning tasks, and proposes a few-shot learning-based multi-modality fusion model named COMFUSE to detect COVID-19 rumors in Sina Weibo. It exploits the meta-training methodology to empower the model to adapt to new events with few instances, as well as fully utilizing two modalities including source posts and comments from the online social media to support the detection of rumors. Experiments on our self-collected Weibo dataset and the publicly available PHEME dataset have shown significant improvement on the COVID-19 few-shot rumor detection task and the generalization capacity of the proposed model.

We would like to sincerely thank our colleagues, Professor Xiao-jun Wu, and Jun Sun from Jiangnan University, Professor Jun-yuan Xie, and Chong-jun Wang from Nanjing University, for their kind support and guidance.

Additional Information and Declarations

Competing Interests

Author Contributions

Data Availability

The authors declare there are no competing interests.

Heng-yang Lu and Chenyou Fan conceived and designed the experiments, performed the experiments, analyzed the data, performed the computation work, prepared figures and/or tables, authored or reviewed drafts of the paper, and approved the final draft.

Xiaoning Song and Wei Fang conceived and designed the experiments, authored or reviewed drafts of the paper, and approved the final draft.

The following information was supplied regarding data availability:

Data and code of COMFUSE are available at GitHub: https://github.com/jncsnlp/FSL-Multimodal-Rumor-Detection.git.

The code is available at GitHub: https://github.com/jncsnlp/FSL-Multimodal-Rumor-Detection/tree/main/COMFUSE.

The complete Weibo dataset is available at GitHub: https://github.com/jncsnlp/Sina-Weibo-Rumors-for-few-shot-learning-research.

The PHEME dataset is available at figshare: Zubiaga, Arkaitz; Wong Sak Hoi, Geraldine; Liakata, Maria; Procter, Rob (2016): PHEME dataset of rumours and non-rumours. figshare. Dataset. https://doi.org/10.6084/m9.figshare.4010619.v1).

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
