# Peer review of "A novel few-shot learning based multi-modality fusion model for COVID-19 rumor detection from online social media"

_PeerJ Computer Science, doi:10.7717/peerj-cs.688_

## Round 0.1 · original submission · Major Revisions

This paper tackles a timely and important research topic, rumour detection in social media. While both reviewers value this work, they also highlight important limitations that will require substantial revisions before it can be considered. Most importantly, reviewers highlight the need to (1) make the novelty of the proposed model clearer, (2) use more recent, competitive methods as baseline models, as the ones used are old, (3) improve the writing and context provided in the paper, as some inconsistencies and need for justification/references has been identified by reviewers.

Reviewer 1 ·

Basic reporting

This paper tends to tackle a very important niche in the topic of rumour detection. Although from the methodological perspective, the dataset and the techniques are novel, there are issues regarding the context that need to be clearly addressed:
1- Several places throughout the paper require citation. It is necessary to refer to the manuscripts that support the claimed statements (e.g., line 65,66,71, etc.)
2- Some of the statements are very strong. For instance, despite what is strongly claimed in line 84-85 as early works, there are plenty of high-quality studies focusing on that approach [3][6]. Such a claim could be agreed upon only if it was supported by a comprehensive literature review or set of references.
3- There is an assumption in this paper that rumour is either unverified or false (The first line of the introduction). Rumours are unverified in some context, and they are not accompanied by substantial evidence for at least some group of people [1] thus if we know that a message is false, then it should not be called rumour anymore.
4- The second paragraph of the introduction is not accurate. What is explained as SDQC support classification is in fact the stance detection which along with rumour detection, rumour tracking and veracity detection constitute the rumour resolution system [2].
5- The second and third paragraphs are semantically disconnected and difficult to follow. The earlier paragraph is about the conceptual phases of the rumour resolution system and the latter one is about the methodological approach (with an emphasis on feature extraction) toward rumour detection.
6- One of the major missing pieces is the gap analysis and the problem formulation. What is expected here is a thorough review of the rumour detection literature in a way that directs readers toward the gap. Here the literature review is unrelated to the gap. For instance, there are several studies on early rumour detection [3] which I expected to see here because this paper aims to flag upcoming rumours as soon as possible without spending time to collect data on the same rumour and that is exactly what early rumour detection system does. Another topic that is expected to be investigated in the literature review is cross-domain rumour detection [4][5]. One of the other missing approaches is rumour detection using context-independent features. Such features are independent of a particular incident and could be used across different domains [5].
7- Another shortcoming is the lack of transparency about the data collection process. Questions such as when did you collect the data? how do the readers retrieve the data and get access to the data points? what keywords did you use to build the queries and collect data? are unanswered. Besides, the events (rumours and non-rumours) are expected to be fully described. The current explanation of the events is quite broad and uninformative.
8- When the dataset is introduced (line 293) the term “instance” is used. Does this term refer to a single message (similar to a tweet on Twitter)? Because not all the readers are Weibo users, it would be helpful to show an example of a post on Weibo visually.
9- Based on my understanding the equivalent terms for Twitter’s reply and retweet in Weibo are comment and repost. If that is correct, then why did you decide to regard the retweets in the PHEME dataset as comments and not repost (line 310-311)?
10- How does the annotation by the Sina Weibo community management centre work (line 159)? What kind of labels a post/datapoint may receive?
11- There are some typos and grammatical issues (e.g,. the first column of Table 1)

[1] Nicholas Difonzo and Prashant Bordia. Rumors influence: Toward a dynamic social impact theory of rumor. Frontiers of social psychology. New York, NY, US:
Psychology Press, 2007, pp. 271–295
[2] A. Zubiaga et al. “Detection and Resolution of Rumours in Social Media”. In: ACM Computing Surveys 51.2 (Feb. 2018), pp. 1–36. DOI: 10.1145/3161603
[3] Kwon, S., Cha, M., & Jung, K. (2017). Rumor detection over varying time windows. PloS one, 12(1), e0168344.
[4] Sicilia, R., Merone, M., Valenti, R., Cordelli, E., D’Antoni, F., De Ruvo, V., ... & Soda, P. (2018, December). Cross-topic rumour detection in the health domain. In 2018 IEEE International Conference on Bioinformatics and Biomedicine (BIBM) (pp. 2056-2063). IEEE.
[5] Fard, A. E., Mohammadi, M., & van de Walle, B. (2020, June). Detecting Rumours in Disasters: An Imbalanced Learning Approach. In International Conference on Computational Science (pp. 639-652). Springer, Cham.
[6] Sicilia, R., Giudice, S. L., Pei, Y., Pechenizkiy, M., & Soda, P. (2018). Twitter rumour detection in the health domain. Expert Systems with Applications, 110, 33-40.

Experimental design

The proposed method is novel and the experiments are well explained; however, there are two issues regarding the robustness of this study:
1- Based on the experimentation setting (line 311-313), you used 3-fold cross-validation. How come you chose three splits here? Why not 5 or 10? You need to justify your decision.
2- For the PHEME dataset, You decided to use #Ferguson unrest, #Ottawa shooting, #Sydney siege as the training and validation set and #Charlie Hebdo shooting, #Germanwings plane crash for the testing (line 303-307). Like the previous point, this decision is not justified as well. One quick fix for both is to do sensitivity analysis by running new experiments. For the k-fold cross-validation issue, this means to run the new experiments when k=3,5, 10 and show to what degree the results change by increasing the number of splits. For the second issue, it means to use different datasets for training-validation and test and show how much the results are dependent on the choice of train-validation-test sets.
Additionally and as I explained before, a coherent chain of related work, research gap, and research questions are absent in this paper. Hence although the experiments and few-shot learning based approach toward rumour detection is very well explained, they are not based on a crystal clear motivation (which comes from an in-depth literature review and subsequent gap identification)

Validity of the findings

The methodological aspect of this study is quite novel and tends to address a very important challenge in automatic rumour detection systems.

·

Basic reporting

See General Comments for the Author.

Experimental design

See General Comments for the Author.

Validity of the findings

See General Comments for the Author.

Additional comments

The spread of rumors will cause the panic of the public and place considerable losses on the economy and other aspects of society. To solve the rumor detection problem on social media, the authors proposed a few-shot learning-based multi-modality fusion model named COMFUSE, including text embeddings modules with pre-trained BERT model, feature extraction module with multilayer Bi-GRUs, multi-modality feature fusion module with a fusion layer, and meta-learning based few-shot learning paradigm for rumor detection. Although the writing is unambiguous, this paper lacks sufficient experiments to verify its contribution. Some concerns are listed as follows:

1. The authors should illustrate the innovation of the proposed model. The modules used in this paper are all based on existing models such as BERT, Bi-GRUs, without any innovative technologies proposed in this paper.

2. The latest baseline for comparison in this paper was proposed in 2018, the authors need to compare the proposed method with more recent baselines.

3. In many related works, such as Bian et al. 2020, Ma et al. 2019, and Liu et al. 2019 cited in the paper, have a rumor early detection experiment. They use very few tweets posted before the early detection deadline as the training set, the models proposed in these papers are tested on the test set, and good detection effects are also obtained. I think the method proposed in this paper should compare with these methods.

4. This paper uses a pre-training model to improve the accuracy of rumor detection. I wonder if this BERT model can be applied to other methods based on textual content of tweets, and can these methods also be significantly improved, even more than the Bi-GRUs based model proposed in this paper?

5. The author should use experimental results to show that rumor detection results are insensitive to different pad sizes of posts and comments.

6. Do not use the same notation for different definitions in the paper, such as b and T.

7. In Table 1, why are MH370, College entrance exams, …, Rabies COVID-19 relevant, and Zhong Nanshan, Wuhan are irrelevant?

In short, the writing is clear but the model lacks innovation. And this paper lacks sufficient experiments to verify its contribution. I suggest that the paper should be greatly modified to make it more acceptable.

---

## Round 0.2 · accepted · Accept

The current revision addresses all of the reviewer comments from the last round, as also indicated by two of the original reviewers. The paper can now be accepted for publication in its current form.

Reviewer 1 ·

Basic reporting

The authors took good care of my comments in the first round of review about literature review and restructuring the paper and I have no further comments and questions.

Experimental design

I don't have any other comments for this section.

Validity of the findings

I don't have any other comments for this section.

·

Basic reporting

no comment

Experimental design

no comment

Validity of the findings

no comment

Additional comments

The authors have revised the manuscript as suggested. I have no further comments and recommend accepting this paper.

---

## Author Rebuttal · Round 0.2

**School of Artificial Intelligence and Computer Science**
Jiangnan University
1800 Lihu Avenue
Wuxi, Jiangsu, China                                              July 7th, 2021

Dear Editors and Reviewers,

We sincerely thank the reviewers for their generous comments on the manuscript and have edited the manuscript to address their concerns. We accept all the suggestions and have responded all the questions. We have carefully revised this manuscript and followings are point-by-point responses to all the comments.

We believe that the manuscript is now suitable for publication in PeerJ Computer Science.

Dr. Heng-yang Lu
Department of Computer Science and Technology

On behalf of all authors.

*Reviewer 1 (Anonymous)*

*Basic reporting*

*This paper tends to tackle a very important niche in the topic of rumour detection. Although from the methodological perspective, the dataset and the techniques are novel, there are issues regarding the context that need to be clearly addressed:*

*1- Several places throughout the paper require citation. It is necessary to refer to the manuscripts that support the claimed statements (e.g., line 65,66,71, etc.)*

Agreed, thank you for your suggestions. The statement about the definition of rumor comes from (Allport et al., 1947), we have referred to the manuscript to support the claimed statements in revisions.

*2- Some of the statements are very strong. For instance, despite what is strongly claimed in line 84-85 as early works, there are plenty of high-quality studies focusing on that approach [3][6]. Such a claim could be agreed upon only if it was supported by a comprehensive literature review or set of references.*

Agreed. We have re-written the introduction and added the literature review section. In the introduction section, we introduce previous works from the perspectives of macro-level (Castillo et al., 2011; Yang et al., 2012; Kwon et al. 2013; Jin et al., 2014; Wu et al., 2015) and micro-level (Sicilia et al., 2017; Sicilia et al., 2018 (a)), more high-quality studies were introduced. In the literature review section, we also introduced these hand-crafted feature-based early works in detail (Castillo et al., 2011; Yang et al., 2012; Kwon et al. 2013; Wu et al., 2015; Liu et al., 2015; Zhao et al., 2015; Jin et al., 2016; Mohammad et al., 2017; Kwon et al., 2017).

*3- There is an assumption in this paper that rumour is either unverified or false (The first line of the introduction). Rumours are unverified in some context, and they are not accompanied by substantial evidence for at least some group of people [1] thus if we know that a message is false, then it should not be called rumour anymore.*

Thank you for your comments. The first line of the introduction is the definition of rumor from early social psychology literature (Allport et al., 1947). We have introduced a more recent definition from DiFonzo [1], which defined rumor as unverified and instrumentally relevant information statements in circulation that arise in contexts of ambiguity and that function primarily to help people make sense and manage threats in revisions.

*4- The second paragraph of the introduction is not accurate. What is explained as SDQC support classification is in fact the stance detection which along with rumour detection, rumour tracking and veracity detection constitute the rumour resolution system [2].*

Agreed, the SDQC support classification and veracity prediction belong to the SemEval-2017 Task 8: RumourEval, we have deleted this paragraph.

*5- The second and third paragraphs are semantically disconnected and difficult to follow. The earlier paragraph is about the conceptual phases of the rumour resolution system and the latter one is about the methodological approach (with an emphasis on feature extraction) toward rumour detection.*

Agreed. We have heavily modified the introduction section and added the literature review section before describing the proposed models to alleviate the disconnection between the introduction and methodological approach. In the introduction section, we first introduce the existing works of rumor detection, then we introduce the motivation of our work, which aims to detect rumors about the emergent event like COVID-19. This problem is different from traditional rumor detection tasks, such that the events in the test set have never occurred before and do not appear in the training set. **In other words, the events in training sets and in testing sets are non-overlapping with each other.** Only very few labeled data (e.g. 1/3/5) of these events are available to adapt tasks, which previous supervised learning-based methods by setting time windows are not suitable in this situation. The cross-topic methods have discussed this problem, and concluded that to obtain good results in cross-topic detection, at least 80% of the test topic knowledge should be included in the training set. Therefore, existing works have huge difficulty in rumor detection for emergent events like COVID-9 with very few labeled data. For these reasons, we introduce the reasonableness and necessity of utilizing few-shot learning methods in this paper. In the literature review section, we comprehensively introduced related works in rumor detection, rumor detection at an early stage, and few-shot learning.

*6- One of the major missing pieces is the gap analysis and the problem formulation. What is expected here is a thorough review of the rumour detection literature in a way that directs readers toward the gap. Here the literature review is unrelated to the gap. For instance, there are several studies on early rumour detection [3] which I expected to see here because this paper aims to flag upcoming rumours as soon as possible without spending time to collect data on the same rumour and that is exactly what early rumour detection system does. Another topic that is expected to be*

*investigated in the literature review is cross-domain rumour detection [4][5]. One of the other missing approaches is rumour detection using context-independent features. Such features are independent of a particular incident and could be used across different domains [5].*

Agreed. Thank you for your constructive suggestions. We have added the literature review section before the methodological approach section. In this section, we firstly introduce related works about rumor detection. Then, we introduced related works in early-stage rumor detection. We review this part with related works from the perspectives of applying time windows on the same rumor (Kwon et al., 2017; Ma et al., 2019; Bian et al., 2020) and detecting rumors under cross-topic situations (Sicilia et al., 2018 (b); Fard et al., 2020). Finally, we introduce related works about few-shot learning, which was related to the techniques proposed in this paper.

*7- Another shortcoming is the lack of transparency about the data collection process. Questions such as when did you collect the data? how do the readers retrieve the data and get access to the data points? what keywords did you use to build the queries and collect data? are unanswered. Besides, the events (rumours and non-rumours) are expected to be fully described. The current explanation of the events is quite broad and uninformative.*

Agreed. We have added more details and illustrations in the Data collection part. (1) the official Sina Weibo community management center displays all the posts which are labeled as rumors, we design a web crawl to crawl all these rumors, dating from May 2012 to December 2020 and then collect selected events from these data. (2) we have shared this dataset and given an access link in the revision manuscript, the detailed posted date of each instance can also be checked in the publicly available repository (https://service.account.weibo.com/). (3) the event names displayed in Table 1 are used as searching keywords (the original names we used are in Chinese, here we have translated them to English in the manuscript). We have added this illustration in revisions. (4) all the descriptions of selected events have been fully explained in the revision.

*8- When the dataset is introduced (line 293) the term "instance" is used. Does this term refer to a single message (similar to a tweet on Twitter)? Because not all the readers are Weibo users, it would be helpful to show an example of a post on Weibo visually.*

Thank you for your suggestions. In the problem setting, we have given the definition of instance used in this paper. An instance $(x_i, y_i)$, $x_i = [m_i, c_i]$, where $x_i$ is a full microblog, $m_i$ refers to the text content (post) of the $i$-th full microblog, and $c_i = [c_{i1}, c_{i2}, ..., c_{il}]$ consists of the $l$ comments of the $i$-th microblog. We regard $m_i$ and $c_i$ as two modalities. $y_i$ is the label of the

*i*-th instance, which indicates whether the *i*-th instance belongs to rumor or not. All these data are collected during the data collection process, and represented as one line in the dataset. We have also added an example of the Weibo page for visualization in Fig. 1 and illustrates each part in the revision manuscript.

*9- Based on my understanding the equivalent terms for Twitter's reply and retweet in Weibo are comment and repost. If that is correct, then why did you decide to regard the retweets in the PHEME dataset as comments and not repost (line 310-311)?*

Yes, the reply and retweet in Twitter are similar to the comment and repost in Sina Weibo. In the pheme dataset, we use the reply as comments, this was a mistake in the writing.

*10- How does the annotation by the Sina Weibo community management centre work (line 159)? What kind of labels a post/datapoint may receive?*

Thank you for your question. The workflow of the rumor judgement by the official Sina Weibo community management center is similar to the process of court ruling. The final judgement by the official platform (on the top of Fig. 2) comes from both reported reasons from other users (on the bottom left) and explanations from the posted user (on the bottom right). Once the post is labeled as a rumor, a 'Fake post' sign would appear on the posted page, as Fig. 1 shows. We have added explanations and Fig.1, 2 in the revision manuscript.

*11- There are some typos and grammatical issues (e.g,. the first column of Table 1)*

Thank you for pointing out this issue, we have corrected on typo in the first column in Table 1, events about *MH370, College entrance exams, etc.* are COVID-19 irrelevant, and events about *Zhong Nanshan, Wuhan, etc.* are COVID-19 relevant. Besides, we have carefully revised our manuscript and polished the language.

*The proposed method is novel and the experiments are well explained; however, there are two issues regarding the robustness of this study:*

*1- Based on the experimentation setting (line 311-313), you used 3-fold cross-validation. How come you chose three splits here? Why not 5 or 10? You need to justify your decision.*

Thank you for your question. In supervised learning, the n-fold cross-validation means to divide the dataset into n folds according to the number of instances in the dataset, use one fold for test and the rest for training. Because the number of instances is more than n, so we can see 5-fold/10-fold cross-validations in many supervised learning experiments. However, this paper conduct rumor detection with few-shot learning, the cross-validation strategy is different from that used in supervised learning. The divide of the training set in few-shot learning is based on distinct events. The number of n-fold cross-validation depends on the number of events in the test set. We take the Weibo dataset for example. We define the few-shot rumor detection for the Weibo dataset as 2-way 3-event 5-shot 9-query. In the Weibo dataset, we have 3 COVID-19 relevant events to be detected with only a few labeled data. The number of the event in the definition is determined by the number of new events in the test dataset, so it is 3-event. The number of ways indicates the number of labels, which are rumor and non-rumor. With this definition, during the few-shot learning training process, every training epoch will sample 3 different events in the training set, for each event, 5 rumor instances and 5 non-rumor instances will be sampled for training. According to the few-shot learning setting, we guarantee that all events in the training sets should NOT appeared in the testing sets, and vice versa, to avoid the leakage of event information and guarantee that we are testing on complete novel events. We also assume that the number of events in the training set should be no less than the number of events in the test set to ensure the model capacity for adapting to new events. According to our assumption and task settings, we split our Weibo dataset to 3 events (COVID-19 relevant) for testing, and 11 events (COVID-19 irrelevant) for training. We fix the 3 events (COVID-19 relevant) for testing, and construct 3 folds for "cross-validation" over 11 training events (COVID-19 irrelevant) to guarantee that each fold has more than 3 events. We have also added these explanations in revisions.

*2- For the PHEME dataset, You decided to use #Ferguson unrest, #Ottawa shooting, #Sydney siege as the training and validation set and #Charlie Hebdo shooting, #Germanwings plane crash for the testing (line 303-307). Like the previous point, this decision is not justified as well. One quick fix for both is to do sensitivity analysis by running new experiments. For the k-fold cross-validation issue, this means to run the new experiments when k=3,5, 10 and show to what degree the results change by increasing the number of splits. For the second issue, it means to use different datasets for training-validation and test and show how much the results are dependent on the choice of train-validation-test sets.*

*Additionally and as I explained before, a coherent chain of related work, research gap, and research questions are absent in this paper. Hence although the experiments and few-shot learning based approach toward rumour detection is very well explained, they are not based on a*

*crystal clear motivation (which comes from an in-depth literature review and subsequent gap identification)*

Thank you for your comments and questions. The followings are point-to-point responses. (1) This paper aims to propose a few-shot learning rumor detection model for detecting rumors about new events based on historical data. We use the publicly available PHEME dataset to show the generality of our work. However, PHEME is not designed for few-shot learning, it only contains 5 events. We choose #Charlie Hebdo shooting, #Germanwings plane crash as the events in the test set because these 2 events have happened later than the previous 3 events (#Ferguson unrest, #Ottawa shooting, #Sydney siege). This is to guarantee the problem setting of few-shot learning rumor detection. (2) The reason why we choose 3-folds has been explained above, please check it. This is mainly because the cross-validation in few-shot learning is different from that in supervised learning. (3) Actually, we have chosen different datasets. We take the PHEME dataset for example. In few-shot learning, the test set contains all the instances that belong to new events, which are all the instances about #Charlie Hebdo shooting and #Germanwings plane crash. We construct 3-fold datasets to train different few-shot learning models to show the stability of our proposed models. The split 0, split 1 and split 2 contains instances about (#Ferguson unrest, #Ottawa shooting), (#Ottawa shooting, #Sydney siege), (#Ferguson unrest, #Sydney siege) respectively. (4) Agreed, we indeed missed the literature review in the previous manuscript, we have responded in the previous comments and added the literature review section in revisions. Thank you for your suggestion.

*Validity of the findings*
*The methodological aspect of this study is quite novel and tends to address a very important challenge in automatic rumour detection systems.*

Thank you.

*Reviewer 2 (Tian Bian)*
*Basic reporting*
*See General Comments for the Author.*
*Experimental design*
*See General Comments for the Author.*
*Validity of the findings*
*See General Comments for the Author.*

*Comments for the Author*

*The spread of rumors will cause the panic of the public and place considerable losses on the economy and other aspects of society. To solve the rumor detection problem on social media, the authors proposed a few-shot learning-based multi-modality fusion model named COMFUSE, including text embeddings modules with pre-trained BERT model, feature extraction module with multilayer Bi-GRUs, multi-modality feature fusion module with a fusion layer, and meta-learning based few-shot learning paradigm for rumor detection. Although the writing is unambiguous, this paper lacks sufficient experiments to verify its contribution. Some concerns are listed as follows:*

*1. The authors should illustrate the innovation of the proposed model. The modules used in this paper are all based on existing models such as BERT, Bi-GRUs, without any innovative technologies proposed in this paper.*

Thank you for your comments. We have completely rewritten the introduction section to show the motivation and innovation of this work. The motivation of our work is to detect rumors about the emergent event like COVID-19. This problem is different from traditional rumor detection tasks, in which the event in the test set has never occurred before and does not appear in the training set. Only an extreme few labeled data (e.g. 1/3/5) of these events are available to task adaptation, this task is different from previous supervised learning-based early-stage rumor detection task. For example, popular models such as GAN-GRU (Ma et al., 2019) and BiGCN (Bian et al., 2020), they can also support early-stage rumor detection. They use early posts whose posted time before a predefined delay to train and predict, and all the events can be found in both the training set and test set. This paper focuses on the cross-topic rumor detection task (Sicilia et al., 2018 (b)), previous work shows that at least 80% of the test topic knowledge should be included in the training set to obtain a good result. Which is a challenging task when the number of labeled new events is scarce. For these reasons, we introduce the reasonableness and necessity of utilizing few-shot learning methods in this paper. We propose a rumor detection model with few-shot learning. The BERT and Bi-GRUs are parts of the model, they offer the features for task adaptation in the few-shot learning procedure.

*2. The latest baseline for comparison in this paper was proposed in 2018, the authors need to compare the proposed method with more recent baselines.*

Agreed, although these models achieved state-of-the-art performance in supervised-learning settings (Ma et al., 2019; Bian et al., 2020), we found that they could only provide below-average performance in the cross-topic/few-shot rumor detection setting. Specifically, these SOTA models

indeed achieved great performance when the events (topics) appear in both the training set and test set, and the performance has a decline when the events discussed in the test set have not appeared in the training set. We have reported the performance of GAN-GRU-early (Ma et al., 2019), BiGCN (Bian et al., 2020) on both datasets in Table 3 and 4. Overall, we found that they under-perform our approach due to the scarce instances related to the emergent events in the test set being fed to the training process. These supervised-based baselines have shown their limitations and are not suitable for the few-shot rumor detection task. (Because we used 3 latest comments as the second modality in COMFUSE, we use the early-stage strategy introduced in these two papers for evaluation.)

*3. In many related works, such as Bian et al. 2020, Ma et al. 2019, and Liu et al. 2019 cited in the paper, have a rumor early detection experiment. They use very few tweets posted before the early detection deadline as the training set, the models proposed in these papers are tested on the test set, and good detection effects are also obtained. I think the method proposed in this paper should compare with these methods.*

Agreed. Thank you for your suggestion. Two models including BiGCN (Bian et al. 2020) and GAN-GRU (Ma et al., 2019) are used for further experiments with their official release codes. However, we found that the source code of Liu's work is not publicly available. In the experimental settings, we used the latest 3 comments of each source post for early-stage evaluation, which is consistent with the comments we used as the second modality in COMFUSE for fair comparisons. Experimental results of two SOTA models GAN-GRU (Ma et al., 2019) and BiGCN (Bian et al., 2020) are reported in Tables 3 and 4 and analyzed in the revisions.

*4. This paper uses a pre-training model to improve the accuracy of rumor detection. I wonder if this BERT model can be applied to other methods based on textual content of tweets, and can these methods also be significantly improved, even more than the Bi-GRUs based model proposed in this paper?*

Thank you for your suggestions. For fair comparisons, we have already used the pretrained BERT model to encode input sequences, and then use DNNs-based methods (SEQ-CNNs and SEQ-Bi-GRUs) to extract features from textual contents for rumor detection. So the results displayed in Tables 3 and 4 of SEQ-CNNs and SEQ-Bi-GRUs are already the experimental results with BERT pretrained model mentioned in your comments. In this way, we guarantee the baseline performance of SEQ-CNNs and SEQ-Bi-GRUs by using the same pretrained BERT embeddings

with our approach, and make completely fair comparison to emphasize the superiority of our proposed method with few-shot learning.

*5. The author should use experimental results to show that rumor detection results are insensitive to different pad sizes of posts and comments.*

Agreed. We have conducted additional experiments to show the rumor detection results with the change of different pad sizes of posts and comments, we take the Weibo dataset as an example. Fig. 9 displays the results of different pad sizes of source posts with a fixed pad size of comments as 32 on the Weibo dataset. Fig. 10 displays the results of different pad sizes of comments with a fixed pad size of source posts as 100. Both $x$-axis refer to the pad size and $y$-axis refers to the accuracy performance. We can observe that the rumor detection results of COMFUSE with different pad sizes of posts and comments vary slightly. For the Weibo dataset, the experimental results reveal that it is relatively better to set the pad size as 100 for posts and 32 for comments, which is consistent with our decision based on the statistics of the length in Fig. 7 and 8.

*6. Do not use the same notation for different definitions in the paper, such as b and T.*

Thank you for your suggestions. We do have this issue when introducing feature extraction with Bi-GRUs. We have carefully revised this part and revised corresponding Figures 1-3 to make sure the same notations only have their own definitions.

*7. In Table 1, why are MH370, College entrance exams, ..., Rabies COVID-19 relevant, and Zhong Nanshan, Wuhan are irrelevant?*

Thank you for pointing out this issue, the first column in Table 1 was written upside down, events about *MH370, College entrance exams, etc.* are COVID-19 irrelevant, and events about *Zhong Nanshan, Wuhan, etc.* are COVID-19 relevant. We have corrected this typo in the revisions.

*In short, the writing is clear but the model lacks innovation. And this paper lacks sufficient experiments to verify its contribution. I suggest that the paper should be greatly modified to make it more acceptable.*

Thank you. We have carefully revised our manuscript with all your comments.

**Allport, G. W, and Postman, L.** The psychology of rumor. 1947.

**Bian T, Xiao X, Xu T, et al.** Rumor Detection on Social Media with Bi-Directional Graph Convolutional Networks[C]. Proceedings of the AAAI Conference on Artificial Intelligence. 2020, 34(01): 549-556.

**Castillo C, Mendoza M, Poblete B.** Information credibility on twitter[C]. Proceedings of the 20th international conference on World wide web. 2011: 675-684.

**DiFonzo, N and Bordia, P.** Rumors influence: Toward a dynamic social impact theory of rumor. Psychology Press. 2011: 271-295.

**Fard AE, Mohammadi M and van de Walle B.** Detecting Rumours in Disasters: An Imbalanced Learning Approach. In International Conference on Computational Science, 2020: 639-652.

**Jin Z, Cao J, Jiang YG and Zhang Y.** News credibility evaluation on microblog with a hierarchical propagation model. In 2014 IEEE International Conference on Data Mining, 2014: 230-239.

**Jin Z, Cao J, Zhang Y, et al.** News verification by exploiting conflicting social viewpoints in microblogs[C]. Proceedings of the AAAI Conference on Artificial Intelligence. 2016, 30(1).

Mohammad S M, Sobhani P and Kiritchenko S. Stance and sentiment in tweets[J]. ACM Transactions on Internet Technology (TOIT), 2017, 17(3): 1-23.

**Kwon S, Cha M, Jung K, Chen W and Wang Y.** Prominent features of rumor propagation in online social media. In 2013 IEEE 13th international conference on data mining, 2013: 1103-1108.

**Kwon, S, Cha, M and Jung, K.** Rumor detection over varying time windows. PloS one. 2017, 12(1), e0168344.

**Liu X, Nourbakhsh A, Li Q, Fang R and Shah S.** Real-time rumor debunking on twitter. In Proceedings of the 24th ACM international on conference on information and knowledge management, 2015: 1867-1870.

**Ma J, Gao W, Wong K F.** Detect rumors on twitter by promoting information campaigns with generative adversarial learning[C]. The World Wide Web Conference. 2019: 3049-3055.

**Sicilia R, Giudice SL, Pei Y, Pechenizkiy M and Soda P.** Health-related rumour detection on Twitter. In 2017 IEEE International Conference on Bioinformatics and Biomedicine (BIBM), 2017: 1599-1606.

**Sicilia R (a), Giudice SL, Pei Y, Pechenizkiy M and Soda P.** Twitter rumour detection in the health domain. Expert Systems with Applications, 2018, 110: 33-40.

**Sicilia R (b), Merone M, Valenti R, Cordelli E, D'Antoni F, De Ruvo V, and Soda P.** Cross-topic rumour detection in the health domain. In 2018 IEEE International Conference on Bioinformatics and Biomedicine (BIBM), 2018: 2056-2063.

**Wu K, Yang S and Zhu KQ.** False rumors detection on sina weibo by propagation structures. In 2015 IEEE 31st international conference on data engineering, 2015: 651-662.

**Yang F, Liu Y, Yu X and Yang M.** Automatic detection of rumor on sina weibo. In Proceedings of the ACM SIGKDD workshop on mining data semantics, 2012: 1-7.

**Zhao Z, Resnick P, Mei Q.** Enquiring minds: Early detection of rumors in social media from enquiry posts[C]. Proceedings of the 24th international conference on world wide web. 2015: 1395-1405.